⟋ | **Open Peer Review** | Clinical Microbiology | Research Article

# Prevalence of vaginal microecological disorder and its association with HPV infections in pregnant women attending prenatal examinations

Jun Shen,[1] Liying Wang,[2,3,4] Wenyu Lin,[5] Dingjie Wang,[2,3,4] Liang Wang,[2,3,4] Jiancui Chen,[1] Pengming Sun[2,3,4]

**ABSTRACT** Vaginal infections are closely associated with human papillomavirus (HPV) infection, yet research on vaginal microecology and HPV mRNA expression in pregnant women remains limited. This study aimed to explore the prevalence of vaginal microecological disorders and their relationship with HPV infection among pregnant women. A cross-sectional study was conducted involving 630 pregnant women who received prenatal care at Fujian Maternity and Child Health Hospital from 2023 to 2024. All participants underwent both the Aptima HPV E6/E7 mRNA assay and vaginal microecological examinations. Clinical characteristics, virological results, and vaginal microecological alterations were analyzed. Of the 630 participants, 241 (38.3%) exhibited vaginal microecology disorders, with aerobic vaginitis (AV, 12.7%) being the most common subtype. A total of 97 (15.4%) participants had received HPV vaccination. The overall high-risk (HR)-HPV E6/E7 mRNA positivity rate was 28.6%, including 4.8%for HPV16/18/45 and 23.8%for the other 11 high-risk genotypes. Age-adjusted multivariate logistic regression analysis identified the following independent risk factors for HR-HPV E6/E7 mRNA positivity: bacterial vaginosis (BV) (aOR = 2.42, 95% CI = 1.25–4.70, $P$ = 0.009), Nugent score ≥7 (aOR= 2.35, 95% CI = 1.21–4.57, $P$ = 0.012), leukocyte esterase positivity (aOR = 1.49, 95% CI = 1.03–2.15, $P$ = 0.036), and lack of HPV vaccination (aOR = 1.99, 95% CI = 1.27–3.12, $P$ = 0.003). Vaginal microecological disorders—particularly BV and leukocyte esterase—are independent risk factors for HR-HPV E6/E7 mRNA positivity. To optimize the management of HPV-infected pregnant individuals, enhanced focus on the prevention and targeted treatment of BV is warranted.

**IMPORTANCE** This study addresses the limited evidence linking vaginal microecology to high-risk human papillomavirus (HR-HPV) infection in pregnancy. Bacterial vaginosis, high Nugent score, and leukocyte esterase positivity were independently associated with HPV infection, whereas HPV vaccination offered protection. These results underscore the value of integrating vaginal microecological assessment and HPV testing into routine prenatal care.

**KEYWORDS** HPV, vaginal microecology, pregnant women

Human papillomavirus (HPV) is the most common sexually transmitted viral infection, particularly among sexually active young women. Most infections are transien t, spontaneous, and asymptomatic (1). However, approximately 10% of women experience persistent high-risk HPV (HR-HPV) infections, placing them at increased risk of cervical cancer and its precursors (2). According to the 2021 IARC report on HPV-associated diseases in China (3), HPV types 16 and 18 account for approximately 69% of cervical cancer cases in China, while types 58, 52, 33, 31, and 45 contribute about 23%, and other high-risk HPV types contribute around 6% of cervical cancers.

**Peer Reviewer** Qun Wang, The First Hospital of Jilin, Changchun, China

Address correspondence to Pengming Sun, fmsun1975@fjmu.edu.cn, or Jiancui Chen, chenjc928@163.com.

Jun Shen and Liying Wang contributed equally to this article. The author order was determined in descending order of seniority.

The authors declare no conflict of interest.

See the funding table on p. 13.

The prevalence of HPV infection is the highest among young, sexually active women under the age of 35 years (4). As women aged 20–35 years make up the majority of the reproductive-age population, they naturally constitute the majority of pregnant women (5). Physiological adaptations during pregnancy, including hormonal fluctuations and immunomodulatory changes, may create a permissive microenvironment for the persistence or progression of HPV infection (6, 7). In general, the data from literature on the incidence of HPV in pregnant women remain inconclusive, varying from 13.0% to 39.4% (8, 9). The prevalence of HPV infection is twice as high in pregnancy as in non-pregnant women (10). Given the increased susceptibility to HPV and high prevalence of HPV infection among pregnant women, there is an urgent need to conduct comprehensive investigations into the HPV infection status in this population.

The pathogenesis of HPV infection principally involves the overexpression of viral oncoproteins. Among them, HPV E6 and E7 have been identified as key oncogene proteins that promote cell proliferation and immortalization (11). The Aptima HPV E6/E7 mRNA test, a novel qualitative nucleic acid amplification assay, detects the combined expression of E6/E7 mRNA from 14 high-risk HPV types, including HPV-16, -18, -31, -33, -35, -39, -45,- 51, -52, -56, -58, -59, -66 and -68, and shows its potential for the identification of HR-HPV infections (12). At present, the primary method of cervical cancer screening during pregnancy is TCT testing. However, the prevalence of HPV mRNA infection during pregnancy has not yet been the subject of a comprehensive investigation.

Vaginal microecology encompasses the vaginal microbiota, vaginal anatomy, immune prevention mechanisms, and endocrine regulatory factors. A healthy vaginal microenvironment serves as the first line of defense against sexually transmitted infections, including HPV infection (13). Once the vaginal flora is disturbed and local immunity is weakened, exogenous microorganisms can invade the female reproductive tract, causing infections and inflammatory processes that increase the risk of reproductive tract diseases and even cancer (14). Vaginal microecological imbalance (e.g., BV) has been demonstrated to increase the risk of HR-HPV infection and adverse birth outcomes for women during pregnancy (15). Such outcomes may include preterm labor, premature rupture of membranes, miscarriage, low birth weight infants, and neonatal infections (16, 17).

Limited data are available on vaginal microecology and HPV mRNA infection among pregnant women. This study aims to investigate the prevalence of vaginal microecological disorder and its association with HPV infections in pregnant women attending prenatal examinations.

## MATERIALS AND METHODS

### Study population

This cross-sectional study recruited pregnant women undergoing cervical screening at Fujian Maternity and Child Health Hospital from 2023 to 2024. Inclusion criteria included the following: (1) age between 20 and 45 years and (2) gestational age between 14 and 24 weeks, with concurrent ThinPrep cytologic test (TCT) and HR-HPV E6/E7 mRNA co-testing. Exclusion criteria included the following: (1) placenta previa, habitual abortion, or high-risk pregnancy; or (2) history of cervical surgery; or (3) hematological system disorders; or (4) recent sexual activity, use of vaginal medication, or douching within 72 hours prior to specimen collection (Fig. 1). All procedures were conducted in accordance with relevant guidelines and regulations, under the oversight of the Ethics Committee.

### Aptima HPV assay

The Aptima HPV assay is a nucleic acid amplification assay that combines target capture, transcription-mediated amplification, and hybridization protection to detect E6/E7

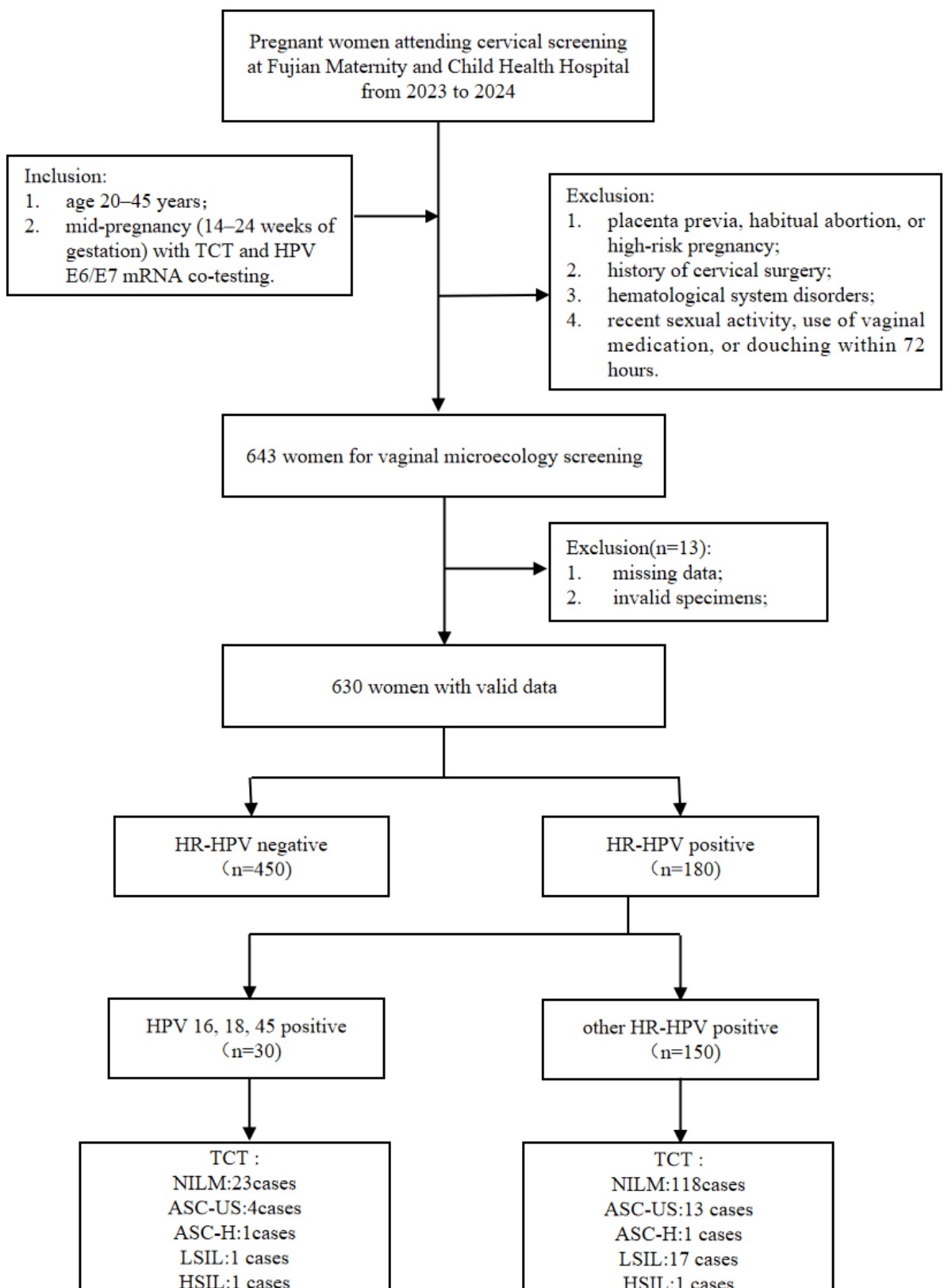

**FIG 1** Flowchart. This flowchart depicts the recruitment of pregnant women (2023–2024, Fujian Maternity and Child Health Hospital) who underwent cervical screening, with 630 participants ultimately enrolled in the study.

mRNA expression of 14 HR-HPV types. The assay is designed to amplify the RNA template under the action of RNA polymerase and reverse transcriptase, producing 100–1,000 copies of the template RNA transcript(s). It can qualitatively detect HPV E6/E7 mRNA from 14 HR-HPV types, namely 16, 18, 31, 33, 35, 39, 45, 51, 52, 56, 58, 59, 66, and 68 in cervical specimens. The Aptima HPV genotyping kit for HPV16, 18/45 E6/E7 mRNA is based on the same principles as the 14 HR-HPV type assays described above and explicitly detects HPV16, 18, and 45. The amplification product is detected using a single-stranded nucleic acid probe complementary to it and incorporating a chemiluminescent marker. The labeled nucleic acid probe specifically binds to the amplification product. The light emitted by the RNA:DNA heterodimer associated with the marker can be quantified using an illuminometer, which detects the photon signal in relative light units (RLU). The ultimate detection outcome is based on the ratio of the signal of this specific analyte to the signal cutoff (S/CO).

## Liquid-based cytology

Cytological specimens were reviewed blind by two experienced cytopathologists, independent of the other assays' results. The results were reported according to the Bethesda 2001 system (18), including atypical squamous cells of undetermined significance (ASC-US), atypical squamous cells that could not exclude high-grade squamous intraepithelial lesion (ASC-H), low-grade squamous intraepithelial lesion (L-SIL), and high-grade squamous intraepithelial lesion (H-SIL). If the diagnosis was different, the cervical sample was evaluated again, and a consistent diagnosis was obtained.

## Vaginal microecology test

A sterile cotton swab was used to collect secretions from the upper third of the vaginal wall. The swabs were then placed in test tubes for immediate analysis. One of the collected swabs was utilized to prepare two slide specimens. The first slide was used for microscopic examination of *Trichomonas vaginalis* and to calculate the Donders score using the physiological saline wet-mount method. The second slide was employed to assess the density and diversity of vaginal flora, determine the Nugent score, and grade *Lactobacillus*. This slide was examined under oil-immersion microscopy after drying, fixation, and Gram staining. Vaginal cleanliness was classified according to the National Clinical Laboratory Practice Guidelines (18th edition): grades I–II were considered normal, while grades III–IV were classified as abnormal. The Nugent score was calculated by evaluating the quantities of *Lactobacillus* morphotypes (scored 0–4), *Gardnerella vaginalis* morphotypes (scored 0–4), and *Mobiluncus* morphotypes (scored 0–2). Leukocyte esterase (LE), neuraminidase (SNA), and hydrogen peroxide ($H_2O_2$) were detected using the bPR-2014A vaginitis automatic analyzer and its detection kit (Master Biotechnology Co., Ltd, Jiangsu, China). Vaginal pH was measured using colorimetric pH strips with a range of 3.8–5.4; a pH <4.5 was considered normal, while a pH ≥4.5 was deemed abnormal.

Diagnostic evaluations were performed based on the Vaginal Microecology Evaluation System (version 2016) (19), as follows: (i) Normal vaginal microecology: density and diversity grades were II–III, the dominant bacteria were Lactobacilli, no pathogenic microorganisms were detected, vaginal pH value was 3.8–4.5, Lactobacilli grade was I–IIa, and LE and SNA were negative. (ii) BV: Nugent score ≥7 points. (iii) Vulvovaginal Candidiasis (VVC): fungal spores or pseudohyphae could be found microscopically under oil. (iv) AV: according to the clinical manifestations and micro-scopic Donders score ≥3 points. (v) Trichomoniasis (TV): a large number of white blood cells and active Trichomoniasis under the microscope. (vi) Mixed vaginitis: two or more types of vaginitis existing at the same time. All laboratory procedures were conducted according to the manufacturer's instructions.

## Statistical analyses

SPSS 22.0 was used for statistical analysis and graphing. Qualitative data were described using frequencies and percentages. Group comparisons were conducted using the chi-square test, Fisher's exact test, or univariate logistic regression analysis. Variables with statistically significant results in univariate analysis were further assessed using multivariate logistic regression analysis, and the odds ratios (ORs) and 95% confidence intervals (CIs) were reported. Statistical tests were two-sided, and $P < 0.05$ was considered statistically significant.

## RESULTS

### Characteristics of the study population

Table 1 summarizes the demographic and clinical characteristics of the study cohort ($N = 630$). The average age was 30.28 ± 4.023 years; educational level was evenly distributed, with 315 women (50%) holding a bachelor's degree or higher and 315 (50%) having a college diploma or lower. Among participants, 533 (84.6%) were unvaccinated against HPV, while 97 (15.4%) had received HPV vaccination (1.9% bivalent, 6.7% quadrivalent, and 6.8% 9-valent). The Aptima HPV assay identified 30 cases (4.8%) positive for HPV-16/18/45, 150 cases (23.8%) positive for the 11 other HR-HPV genotypes, and 450 cases (71.4%) tested negative. Regarding TCT results, 568 cases (90.1%) were categorized as NILM, 32 cases (5.1%) as ASC-US, 2 cases (0.3%) as ASC-H, 25 cases (4.0%) as LSIL, 2 cases (0.3%) as HSIL, and 1 case (0.2%) showed AGC. Vaginal microecological testing revealed 389 (61.7%) participants with normal status and 241 (38.3%) with abnormal status, including 80 (12.7%) with AV, 65 (10.3%) with VVC, 57 (9.0%) with CV, and 39 (6.2%) with BV.

### Age-stratified distribution of HR-HPV types in pregnant populations

To evaluate age-related differences in HPV genotype distribution, chi-square tests were performed across three infection categories (Table 2). For overall HR-HPV infection, 171 (30.2%) of women in the <35 years group tested positive, compared to 9 (14.3%) in the ≥35 years group, with a significant difference ($P = 0.008$). For the 11 other HR-HPV types, 142 (25.0%) of the <35 years group were positive, compared to 8 (12.7%) of the ≥35 years group ($P=0.029$). Regarding HR-HPV 16/18/45, 29 (5.1%) of the <35 years group tested positive, compared to 1 (1.6%) of the ≥35 years group; the difference was not statistically significant ($P =0.350$).

### Association between vaginal microecological alterations and HR-HPV positivity in pregnant women

To further investigate the association between vaginal microecological indicators and HR-HPV positivity in pregnant women, we performed unadjusted and age-adjusted logistic regression analyses (Table 3). The following factors were identified as significant correlates of HR-HPV positivity: BV (adjusted OR = 2.42, 95% CI = 1.25–4.70, $P = 0.009$), Nugent score ≥7 (adjusted OR = 2.35, 95% CI = 1.21–4.57, $P = 0.012$), leukocyte esterase positivity (adjusted OR = 1.49, 95% CI = 1.03–2.15, $P = 0.036$), and unvaccinated status (adjusted OR = 1.99, 95% CI = 1.27–3.12, $P = 0.003$). In contrast, parameters including vaginal pH and flora density showed no significant association with HR-HPV positivity in the adjusted models (all $P > 0.05$). These findings indicate that specific vaginal microecological disruptions (BV, elevated Nugent scores, and leukocyte esterase positivity) and lack of HPV vaccination are linked to HR-HPV positivity in this pregnant cohort.

We further evaluated the association between vaginal microecological parameters and TCT results in 180 HR-HPV-positive pregnant women (Table 4). No statistically significant associations were observed between TCT results and vaginal microecological parameters, including vaginal pH, flora density, bacterial diversity, VVC, AV, BV, CV, Nugent score, and leukocyte esterase (all $P > 0.05$).

**TABLE 1** Socio-demographic characteristics of participants ($N = 630$)[a]

| Patient characteristics | Case count ($N = 630$) | Percent (%) |
|---|---|---|
| Age (years) | $30.28 \pm 4.023$ | /[b] |
| Educational level | | |
| Bachelor's degree or above | 315 | 50% |
| College diploma or below | 315 | 50% |
| HPV vaccination status | | |
| Unvaccinated | 533 | 84.6% |
| Vaccinated | 97 | 15.4% |
| 2V-HPV | 12 | 1.9% |
| 4V-HPV | 42 | 6.7% |
| 9V-HPV | 43 | 6.8% |
| Vaginal microecology test | | |
| Normal | 389 | 61.7% |
| Abnormal | 241 | 38.3% |
| BV | 39 | 6.2% |
| VVC | 65 | 10.3% |
| AV | 80 | 12.7% |
| CV | 57 | 9.0% |
| HPV | | |
| Negative | 450 | 71.4% |
| HPV-16/18/45 positive | 30 | 4.8% |
| Other 11 types of HR-HPV positive | 150 | 23.8% |
| TCT | | |
| NILM | 568 | 90.1% |
| ASC-US | 32 | 5.1% |
| ASC-H | 2 | 0.3% |
| LSIL | 25 | 4.0% |
| HSIL | 2 | 0.3% |
| AGC | 1 | 0.2% |

[a]2V-HPV: bivalent human papillomavirus; 4V-HPV: quadrivalent human papillomavirus; 9V-HPV: nine-valent human papillomavirus; HPV: human papillomavirus; NILM: negative for intraepithelial lesion or malignancy; ASC-US: atypical squamous cells of undetermined significance; ASC-H: atypical squamous cells cannot exclude high-grade squamous intraepithelial lesion; LSIL: low-grade squamous intraepithelial lesion; HSIL: high-grade squamous intraepithelial lesion; AGC: atypical glandular cells.
[b]/ (not applicable).

## Association between vaginal microecological alterations and non-16/18/45 HR-HPV positivity in pregnant women

Moreover, Table 5 delineates the associations between vaginal microenvironment parameters and non-16/18/45 HR-HPV genotypes. The results confirm that abnormal flora density (adjusted OR = 2.18, 95% CI = 1.08–4.40, $P = 0.030$), AV positivity (adjusted

**TABLE 2** Distribution of HR-HPV types by age group ($N = 630$)[a]

| Characteristics, n (%) | <35 ($n = 567$) | ≥35 ($n = 63$) | X2 | P |
|---|---|---|---|---|
| HR-HPV | | | **7.000** | **0.008** |
| Negative ($n = 450$) | 396 (69.8%) | 54 (85.7%) | | |
| Positive ($n = 180$) | 171 (30.2%) | 9 (14.3%) | | |
| Other 11 types of HR-HPV | | | **4.764** | **0.029** |
| Negative ($n = 480$) | 425 (75.0%) | 55 (87.3%) | | |
| Positive ($n = 150$) | 142 (25.0%) | 8 (12.7%) | | |
| HR-HPV16/18/45 | | | 1.662 | 0.350 |
| Negative ($n = 600$) | 538 (94.9%) | 62 (98.4%) | | |
| Positive ($n = 30$) | 29 (5.1%) | 1 (1.6%) | | |

[a]Bold text highlights statistically significant differences between groups ($P < 0.05$).

**TABLE 3** Association between vaginal microecological alterations and HR-HPV mRNA positivity in pregnant women (N = 630)[a,b]

| Characteristics, n (%) | HR-HPV negative (n = 450) | HR-HPV positive (n = 180) | Unadjusted | | Age adjusted | |
|---|---|---|---|---|---|---|
| | | | OR (95% CI) | P | OR (95% CI) | P |
| PH | | | | | | |
| Normal (n = 323) | 222 (49.3%) | 101 (56.1%) | Ref. | | | |
| Abnormal (n = 307) | 228 (50.7%) | 79 (43.9%) | 0.76 (0.54–1.08) | 0.125 | 0.79 (0.55–1.11) | 0.177 |
| Cleanliness levels | | | | | | |
| Normal (n = 332) | 232 (51.6%) | 100 (55.6%) | Ref. | | | |
| Abnormal (n = 298) | 218 (48.4%) | 80 (44.4%) | 0.85 (0.60–1.20) | 0.364 | 0.84 (0.60–1.20) | 0.341 |
| Flora density | | | | | | |
| Normal (n = 594) | 429 (95.3%) | 165 (91.7%) | Ref. | | | |
| Abnormal (n = 36) | 21 (4.7%) | 15 (8.3%) | 1.86 (0.93–3.69) | 0.077 | 1.88 (0.94–3.74) | 0.074 |
| Bacterial diversity | | | | | | |
| Normal (n = 620) | 442 (98.2%) | 178 (98.9%) | Ref. | | | |
| Abnormal (n = 10) | 8 (1.8%) | 2 (1.1%) | 0.62 (0.13–2.95) | 0.549 | 0.59 (0.12–2.82) | 0.510 |
| VVC | | | | | | |
| Negative (n = 565) | 402 (89.3%) | 163 (90.6%) | Ref. | | | |
| Positive (n = 65) | 48 (10.7%) | 17 (9.4%) | 0.87 (0.49–1.56) | 0.649 | 0.82 (0.46–1.48) | 0.515 |
| AV | | | | | | |
| Negative (n = 550) | 398 (88.4%) | 152 (84.4%) | Ref. | | | |
| Positive (n = 80) | 52 (11.6%) | 28 (15.6%) | 1.41 (0.86–2.32) | 0.175 | 1.43 (0.87–2.36) | 0.158 |
| BV | | | | | | |
| Negative (n = 591) | 429 (95.3%) | 162 (90.0%) | Ref. | | | |
| Positive (n = 39) | 21 (4.7%) | 18 (10.0%) | 2.27 (1.18–4.37) | **0.014** | 2.42 (1.25–4.70) | **0.009** |
| CV | | | | | | |
| Negative (n = 573) | 407 (90.4%) | 166 (92.2%) | Ref. | | | |
| Positive (n = 57) | 43 (9.6%) | 14 (7.8%) | 0.80 (0.43–1.50) | 0.483 | 0.74 (0.39–1.40) | 0.358 |
| Nugent score | | | | | | |
| Score: 0–3 (n = 532) | 383 (85.1%) | 149 (82.8%) | Ref. | | | |
| Score: 4–6 (n = 59) | 46 (10.2%) | 13 (7.2%) | 0.73 (0.38–1.38) | 0.331 | 0.72 (0.37–1.37) | 0.313 |
| Score: ≥7 (n = 39) | 21 (4.7%) | 18 (10.0%) | 2.20 (1.14–4.25) | **0.019** | 2.35 (1.21–4.57) | **0.012** |
| LBG | | | | | | |
| Negative (n = 506) | 369 (82.0%) | 137 (76.1%) | Ref. | | | |
| Positive (n = 124) | 81 (18.0%) | 43 (23.9%) | 1.43 (0.94–2.17) | 0.094 | 1.43 (0.94–2.17) | 0.098 |
| Clue cells | | | | | | |
| Negative (n = 604) | 434 (96.4%) | 170 (94.4%) | Ref. | | | |
| Positive (n = 26) | 16 (3.6%) | 10 (5.6%) | 1.60 (0.71–3.59) | 0.258 | 1.76 (0.77–3.99) | 0.178 |
| Catalase | | | | | | |
| Negative (n = 140) | 96 (21.3%) | 44 (24.4%) | Ref. | | | |
| Positive (n = 490) | 354 (78.7%) | 136 (75.6%) | 0.84 (0.56–1.26) | 0.396 | 0.88 (0.58–1.33) | 0.552 |
| Sialidase | | | | | | |
| Negative (n = 599) | 430 (95.6%) | 169 (93.9%) | Ref. | | | |
| Positive (n = 31) | 20 (4.4%) | 11 (6.1%) | 1.40 (0.66–2.98) | 0.384 | 1.41 (0.66–3.03) | 0.373 |
| Leukocyte esterase | | | | | | |
| Negative (n = 443) | 328 (72.9%) | 115 (63.9%) | Ref. | | | |
| Positive (n = 187) | 122 (27.1%) | 65 (36.1%) | 1.52 (1.05–2.20) | **0.026** | 1.49 (1.03–2.15) | **0.036** |
| Leukocyte | | | | | | |
| Negative (n = 347) | 251 (55.8%) | 96 (53.3%) | Ref. | | | |
| Positive (n = 283) | 199 (44.2%) | 84 (46.7%) | 1.10 (0.78–1.56) | 0.577 | 1.09 (0.77–1.55) | 0.611 |
| HPV vaccination status | | | | | | |
| Unvaccinated (n = 533) | 394 (87.6%) | 139 (77.2%) | Ref. | | | |
| Vaccinated (n = 97) | 56 (12.4%) | 41 (22.8%) | 2.08 (1.33–3.24) | **0.001** | 1.99 (1.27–3.12) | **0.003** |

[a]pH <4.5 was considered normal, while a pH ≥4.5 was deemed abnormal. Vaginal cleanliness grades I–II were considered normal, while grades III–IV were classified as abnormal. Density and diversity at levels 2 and 3 are considered normal, while levels 1 or 4 are classified as abnormal.
[b]Bold text highlights statistically significant differences between groups (P < 0.05).

**TABLE 4** Association between vaginal microecological parameters and TCT results in 180 HR-HPV-positive pregnant women (N = 180)[a,b]

| Characteristics, n (%) | NILM (N = 141) | ≥ASC US (N = 39) | Unadjusted | | Age adjusted | |
|---|---|---|---|---|---|---|
| | | | OR (95% CI) | P | OR (95% CI) | P |
| PH | | | | | | |
| Normal (n = 101) | 82 (58.2%) | 19 (48.7%) | Ref. | | | |
| Abnormal (n = 79) | 59 (41.8%) | 20 (51.3%) | 1.46 (0.72–2.98) | 0.295 | 1.46 (0.72–2.98) | 0.295 |
| Cleanliness levels | | | | | | |
| Normal (n = 100) | 71 (50.4%) | 29 (74.4%) | Ref. | | | |
| Abnormal (n = 80) | 70 (49.6%) | 10 (25.6%) | 0.35 (0.16–0.77) | **0.009** | 0.35 (0.16–0.76) | **0.009** |
| Flora density | | | | | | |
| Normal (n = 165) | 128 (90.8%) | 37 (94.9%) | Ref. | | | |
| Abnormal (n = 15) | 13 (9.2%) | 2 (5.1%) | 0.53 (0.11–2.47) | 0.420 | 0.54 (0.12–2.50) | 0.429 |
| Bacterial diversity | | | | | | |
| Normal (n = 178) | 140 (99.3%) | 38 (97.4%) | Ref. | | | |
| Abnormal (n = 2) | 1 (0.7%) | 1 (2.6%) | 3.68 (0.23–60.28) | 0.360 | 3.53 (0.21–58.44) | 0.378 |
| VVC | | | | | | |
| Negative (n = 163) | 126 (89.4%) | 37 (94.9%) | Ref. | | | |
| Positive (n = 17) | 15 (10.6%) | 2 (5.1%) | 0.45 (0.10–2.08) | 0.309 | 0.44 (0.10–2.02) | 0.289 |
| AV | | | | | | |
| Negative (n = 152) | 116 (82.3%) | 36 (92.3%) | Ref. | | | |
| Positive (n = 28) | 25 (17.7%) | 3 (7.7%) | 0.39 (0.11–1.36) | 0.138 | 0.38 (0.11–1.34) | 0.134 |
| BV | | | | | | |
| Negative (n = 162) | 126 (89.4%) | 36 (92.3%) | Ref. | | | |
| Positive (n = 18) | 15 (10.6%) | 3 (7.7%) | 0.70 (0.19–2.55) | 0.589 | 0.71 (0.19–2.58) | 0.598 |
| CV | | | | | | |
| Negative (n = 166) | 131 (92.9%) | 35 (89.7%) | Ref. | | | |
| Positive (n = 14) | 10 (7.1%) | 4 (10.3%) | 1.50 (0.44–5.06) | 0.516 | 1.48 (0.44–5.03) | 0.527 |
| Nugent score | | | | | | |
| Score: 0–3 (n = 149) | 115 (81.6%) | 34 (87.2%) | Ref. | | | |
| Score: 4–6 (n = 13) | 11 (7.8%) | 2 (5.1%) | 0.61 (0.13–2.91) | 0.540 | 0.60 (0.13–2.85) | 0.518 |
| Score: ≥7 (n = 18) | 15 (10.6%) | 3 (7.7%) | 0.68 (0.18–2.48) | 0.555 | 0.68 (0.19–2.50) | 0.562 |
| LBG | | | | | | |
| Negative (n = 137) | 105 (74.5%) | 32 (82.1%) | Ref. | | | |
| Positive (n = 43) | 36 (25.5%) | 7 (17.9%) | 0.64 (0.26–1.57) | 0.328 | 0.62 (0.25–1.54) | 0.304 |
| Clue cells | | | | | | |
| Negative (n = 170) | 133 (94.3%) | 37 (94.9%) | Ref. | | | |
| Positive (n = 10) | 8 (5.7%) | 2 (5.1%) | 0.90 (0.18–4.41) | 0.895 | 0.92 (0.19–4.53) | 0.916 |
| Catalase | | | | | | |
| Negative (n = 44) | 35 (24.8%) | 9 (23.1%) | Ref. | | | |
| Positive (n = 136) | 106 (75.2%) | 30 (76.9%) | 1.10 (0.48–2.54) | 0.822 | 1.11 (0.48–2.57) | 0.806 |
| Sialidase | | | | | | |
| Negative (n = 169) | 130 (92.2%) | 39 (100.0%) | | . | | . |
| Positive (n = 11) | 11 (7.8%) | 0 (0.0%) | NA[c] | . | NA | . |
| Leukocyte esterase | | | | | | |
| Negative (n = 115) | 87 (61.7%) | 28 (71.8%) | Ref. | | | |
| Positive (n = 65) | 54 (38.3%) | 11 (28.2%) | 0.63 (0.29–1.37) | 0.248 | 0.62 (0.29–1.36) | 0.232 |
| Leukocyte | | | | | | |
| Negative (n = 96) | 69 (48.9%) | 27 (69.2%) | Ref. | | | |
| Positive (n = 84) | 72 (51.1%) | 12 (30.8%) | 0.43 (0.20–0.91) | **0.027** | 0.43 (0.20–0.91) | **0.027** |
| HPV vaccination status | | | | | | |
| Unvaccinated (n = 139) | 106 (75.2%) | 33 (84.6%) | Ref. | | | |
| Vaccinated (n = 41) | 35 (24.8%) | 6 (15.4%) | 0.55 (0.21–1.42) | 0.218 | 0.54 (0.21–1.39) | 0.200 |

[a]TCT: ThinPrep cytology test; ≥ASC-US includes: ASC-US, ASC-H, LSIL, HSIL, and AGC.
[b]Bold text highlights statistically significant differences between groups (P < 0.05).
[c]NA (not applicable).

**TABLE 5** Association between vaginal microecological alterations and non-16/18/45 HR-HPV positivity in pregnant women ($N = 630$)[a]

| Characteristics, n (%) | Non-16/18/45 HR-HPV negative ($n = 480$) | Non-16/18/45 HR-HPV positive ($n = 150$) | Unadjusted | | Age adjusted | |
|---|---|---|---|---|---|---|
| | | | OR (95% CI) | P | OR (95% CI) | P |
| PH | | | | | | |
| Normal ($n = 323$) | 237 (49.4%) | 86 (57.3%) | Ref. | | | |
| Abnormal ($n = 307$) | 243 (50.6%) | 64 (42.7%) | 0.73 (0.50–1.05) | 0.089 | 0.75 (0.52–1.10) | 0.140 |
| Cleanliness levels | | | | | | |
| Normal ($n = 332$) | 251 (52.3%) | 81 (54.0%) | Ref. | | | |
| Abnormal ($n = 298$) | 229 (47.7%) | 69 (46.0%) | 0.93 (0.65–1.35) | 0.715 | 0.92 (0.64–1.34) | 0.677 |
| Flora density | | | | | | |
| Normal ($n = 594$) | 458 (95.4%) | 136 (90.7%) | Ref. | | | |
| Abnormal ($n = 36$) | 22 (4.6%) | 14 (9.3%) | 2.14 (1.07–4.30) | **0.032** | 2.18 (1.08–4.40) | **0.030** |
| Bacterial diversity | | | | | | |
| Normal ($n = 620$) | 472 (98.3%) | 148 (98.7%) | Ref. | | | |
| Abnormal ($n = 10$) | 8 (1.7%) | 2 (1.3%) | 0.80 (0.17–3.80) | 0.776 | 0.75 (0.16–3.60) | 0.722 |
| VVC | | | | | | |
| Negative ($n = 565$) | 430 (89.6%) | 135 (90.0%) | Ref. | | | |
| Positive ($n = 65$) | 50 (10.4%) | 15 (10.0%) | 0.96 (0.52–1.76) | 0.884 | 0.89 (0.48–1.64) | 0.702 |
| AV | | | | | | |
| Negative ($n = 550$) | 426 (88.8%) | 124 (82.7%) | Ref. | | | |
| Positive ($n = 80$) | 54 (11.3%) | 26 (17.3%) | 1.65 (0.99–2.75) | 0.053 | 1.69 (1.01–2.83) | **0.044** |
| BV | | | | | | |
| Negative ($n = 591$) | 457 (95.2%) | 134 (89.3%) | Ref. | | | |
| Positive ($n = 39$) | 23 (4.8%) | 16 (10.7%) | 2.37 (1.22–4.62) | **0.011** | 2.58 (1.31–5.09) | **0.006** |
| CV | | | | | | |
| Negative ($n = 573$) | 435 (90.6%) | 138 (92.0%) | Ref. | | | |
| Positive ($n = 57$) | 45 (9.4%) | 12 (8.0%) | 0.84 (0.43–1.63) | 0.609 | 0.76 (0.39–1.50) | 0.432 |
| Nugent score | | | | | | |
| Score: 0–3 ($n = 532$) | 410 (85.4%) | 122 (81.3%) | Ref. | | | |
| Score: 4–6 ($n = 59$) | 47 (9.8%) | 12 (8.0%) | 0.86 (0.44–1.67) | 0.652 | 0.84 (0.43–1.65) | 0.620 |
| Score: ≥7 ($n = 39$) | 23 (4.8%) | 16 (10.7%) | 2.34 (1.20–4.57) | **0.013** | 2.54 (1.28–5.02) | **0.007** |
| LBG | | | | | | |
| Negative ($n = 506$) | 396 (82.5%) | 110 (73.3%) | Ref. | | | |
| Positive ($n = 124$) | 84 (17.5%) | 40 (26.7%) | 1.71 (1.11–2.64) | **0.014** | 1.71 (1.11–2.65) | **0.015** |
| Clue cells | | | | | | |
| Negative ($n = 604$) | 462 (96.3%) | 142 (94.7%) | Ref. | | | |
| Positive ($n = 26$) | 18 (3.8%) | 8 (5.3%) | 1.45 (0.62–3.40) | 0.397 | 1.63 (0.68–3.88) | 0.272 |
| Catalase | | | | | | |
| Negative ($n = 140$) | 101 (21.0%) | 39 (26.0%) | Ref. | | | |
| Positive ($n = 490$) | 379 (79.0%) | 111 (74.0%) | 0.76 (0.50–1.16) | 0.203 | 0.81 (0.53–1.24) | 0.335 |
| Sialidase | | | | | | |
| Negative ($n = 599$) | 459 (95.6%) | 140 (93.3%) | Ref. | | | |
| Positive ($n = 31$) | 21 (4.4%) | 10 (6.7%) | 1.56 (0.72–3.39) | 0.261 | 1.58 (0.72–3.47) | 0.250 |
| Leukocyte esterase | | | | | | |
| Negative ($n = 443$) | 348 (72.5%) | 95 (63.3%) | Ref. | | | |
| Positive ($n = 187$) | 132 (27.5%) | 55 (36.7%) | 1.53 (1.04–2.25) | **0.033** | 1.48 (1.00–2.19) | **0.048** |
| Leukocyte | | | | | | |
| Negative ($n = 347$) | 270 (56.3%) | 77 (51.3%) | Ref. | | | |
| Positive ($n = 283$) | 210 (43.8%) | 73 (48.7%) | 1.22 (0.84–1.76) | 0.291 | 1.21 (0.84–1.75) | 0.313 |
| HPV vaccination status | | | | | | |
| Unvaccinated ($n = 533$) | 421 (87.7%) | 112 (74.7%) | Ref. | | | |
| Vaccinated ($n = 97$) | 59 (12.3%) | 38 (25.3%) | 2.42 (1.53–3.83) | **<0.001** | 2.30 (1.45–3.66) | **<0.001** |

[a]Bold text highlights statistically significant differences between groups ($P < 0.05$).

OR = 1.69, 95% CI=1.01–2.83, *P* = 0.044), BV positivity (adjusted OR = 2.58, 95% CI = 1.31–5.09, *P* = 0.006), Nugent score ≥7 (adjusted OR = 2.54, 95% CI = 1.28–5.02, *P*=0.007), and leukocyte esterase positivity (adjusted OR = 1.48, 95% CI = 1.00–2.19, *P* = 0.048) were associated with non-16/18/45 HR-HPV positivity. In contrast, parameters including vaginal pH, cleansing levels, bacterial diversity, VVC, CV, catalase, and sialidase showed no significant association with the target HR-HPV subtype in adjusted models (all *P* > 0.05).

## Distribution of HPV genotypes in pregnant women with BV

To further examine HR-HPV infection patterns among women with BV, Table 6 shows 18 (46.2%) of BV-positive women tested positive for overall HR-HPV, compared to 162 (27.4%) of BV-negative women. Age-adjusted logistic regression confirmed that BV positivity was associated with significantly higher odds of overall HR-HPV infection (adjusted OR = 2.43, 95% CI = 1.25–4.71, *P* = 0.009). For the 11 other HR-HPV types, 16 (41.0%) of BV-positive women were positive versus 134 (22.7%) of BV-negative women; this association also remained significant after age adjustment (adjusted OR = 2.60, 95% CI = 1.32–5.13, *P* = 0.006). In contrast, no significant correlation was observed between BV status and HR-HPV 16/18/45 positivity: 2 (5.1%) of BV-positive women and 28 (4.7%) of BV-negative women tested positive for these genotypes (adjusted *P* = 0.948).

## DISCUSSION

HPV mixed vaginitis is a prevalent lower genital tract infection in women but has been inadequately studied in pregnant populations. In the present study, 97 (15.4%) participants had received HPV vaccines. The Aptima HPV assay detected 30 (4.8%) cases positive for HPV-16/18/45 and 150 (23.8%) positive for 11 other HR-HPV genotypes. Among vaginal infections, AV had the highest prevalence (12.7%), followed by VVC (10.3%), CV (9.0%), and BV (6.2%). Notably, BV, elevated Nugent score, leukocyte esterase positivity, and HPV-unvaccinated status were identified as risk factors for HPV infection during pregnancy. These findings suggest that vaginal microecological imbalance may serve as an independent risk factor for persistent HPV infection.

A population-based meta-analysis indicates that the overall prevalence of HR-HPV infection among women aged 25–45 years in mainland China is 22.3% (20). In the present study, E6/E7 mRNA detection technology was utilized to ascertain the HR-HPV positivity rate among pregnant women, which was found to be 28.6%. The mRNA positivity rate for the HPV types 16/18/45 was found to be 4.8%. A 9-year study in Fujian Province showed that HPV-52, -58, -16, -39, -51, and -68 were the most prevalent genotypes among women who underwent cervical cancer screening between 2014 and 2022, with HPV 16/18 prevalence of 1.74% (1.71%–1.78%) (21). A population-based study of 63,553 residents in Xiamen, Fujian, reported the prevalence of HPV infection corresponding to vaccine-targeted genotypes: 3.56% (2,264) were positive for the bivalent vaccine-covered types (16/18), 5.89% (3,746) for the quadrivalent vaccine-covered types (6/11/16/18), and 13.64% (8,666) for the nonavalent vaccine-covered types (6/11/16/18/31/33/45/52/58). Consistent with these findings, the low HPV 16/18 infection rate observed in our study, together with the aforementioned epidemiological data, confirms the long-term efficacy of HPV vaccination. This highlights the urgent need to further expand access to the HPV vaccine (22).

Compared with conventional HPV DNA detection in a large cohort (*n* = 15,042) where the HR-HPV positivity rate among reproductive-age women aged 26–35 years was 17.59% (20), our study reported a higher HR-HPV E6/E7 mRNA positivity rate of 28.6%. A recent review highlights that HR-HPV prevalence varies substantially (5.5%–65%) across studies, influenced by factors including maternal age, geographic region, and gestational age (with a gradual increase as pregnancy progresses) (23). Previous findings (24) have shown that HR-HPV E6/E7 mRNA and HPV DNA assays exhibit no significant difference in the detection rate of CIN II/III. Given the relatively high HR-HPV prevalence observed in our pregnant cohort recruited from a provincial tertiary hospital, further multicenter

TABLE 6   Distribution of HPV genotypes and age in pregnant women with BV (N = 630)[a]

| Characteristics, n (%) | BV negative (n = 591) | BV positive (n = 39) | Unadjusted | | Age adjusted | |
|---|---|---|---|---|---|---|
| | | | OR (95% CI) | P | OR (95% CI) | P |
| HR-HPV | | | | | | |
| Negative (n = 450) | 429 (72.6%) | 21 (53.8%) | Ref. | | | |
| Positive (n = 180) | 162 (27.4%) | 18 (46.2%) | 2.27 (1.18–4.37) | **0.014** | 2.43 (1.25–4.71) | **0.009** |
| Other 11 HR-HPV types | | | | | | |
| Negative (n = 480) | 457 (77.3%) | 23 (59.0%) | Ref. | | | |
| Positive (n = 150) | 134 (22.7%) | 16 (41.0%) | 2.37 (1.22–4.62) | **0.011** | 2.60 (1.32–5.13) | **0.006** |
| HR-HPV16/18/45 | | | | | | |
| Normal (n = 600) | 563 (95.3%) | 37 (94.9%) | Ref. | | | |
| Abnormal (n = 30) | 28 (4.7%) | 2 (5.1%) | 1.09 (0.25–4.74) | 0.912 | 1.05 (0.24–4.60) | 0.948 |

[a]Bold text highlights statistically significant differences between groups (P < 0.05).

studies are warranted to systematically characterize HPV infection epidemiology in pregnant populations, with detailed stratification by age and gestational stage to clarify potential variations.

In our study, AV was the most prevalent (12.7%, 80/630), followed by VVC (10.3%, 65/630), and BV had the lowest prevalence (6.2%, 39/630). Studies have shown that the epidemiological characteristics of vaginal inflammation exhibit significant heterogeneity across different geographical regions and population demographics. The prevalence rate of AV ranges from 7.9% to 23.7%, with a lower prevalence rate observed among pregnant women (4.1% to 8.3%) (25). The global prevalence of BV exhibits significant regional heterogeneity. Research by Krauss-Silva et al. demonstrates that the prevalence of BV among black women (32.5%) exceeds that observed among white women (28.1%) (26). The prevalence of BV in pregnant women varies considerably (4.9%–49%) (27). The prevalence of VVC varies significantly among different populations, with rates reaching as high as 29.2% among pregnant women (28). TV has the lowest prevalence rate, with a prevalence rate of 1.7%–4.5% among non-pregnant gynecological outpatients in China and 1.7%–3.2% among pregnant women (29, 30).

Regression analysis identifies BV, elevated Nugent scores, and leukocyte esterase positivity as independent risk factors for HPV infection during pregnancy. The Nugent score is a standardized microscopic scoring system for vaginal secretions, which assesses bacterial composition and serves as the gold standard for laboratory diagnosis of BV. Accumulating evidence supports a strong correlation between BV and HR-HPV infection: BV development is closely linked to vaginal biofilm formation (31), and virulence factors produced by BV-associated bacteria can disrupt the vaginal epithelial mucosal barrier, form protective biofilms to evade host immunity, and thereby facilitate persistent HPV colonization. Specifically, *Gardnerella vaginalis*—a key pathogen in BV—adheres tightly to vaginal epithelial cells to form dense biofilms and secretes vaginal cytolysin, which is hypothesized to inhibit immunoglobulin A (IgA) function in the vaginal mucosal barrier.

Biofilms are defined as surface-attached microbial communities encased in an extracellular polymeric matrix composed of polysaccharides, proteins, and nucleic acids. Their formation also contributes to the high recurrence rate of BV (18, 32). *Gardnerella vaginalis* is responsible for the production of SNA, a substance capable of degrading mucosal protective factors (such as mucin) and inducing vaginal epithelial cell lysis and discharge (33). SNA is an enzyme that cleaves terminal sialic acid residues and is associated with tissue destruction, immune response evasion, bacterial invasion, and nutrient acquisition by bacteria (34). In addition to *Gardnerella* bacteria, other anaerobic bacteria such as *Prevotella*, *Bacteroides*, and *Mobiluncus* have also been shown to produce SNA (35). Anaerobic bacteria have also been demonstrated to produce carcinogens, including nitrosamines, which have been shown to increase susceptibility to viral infection (36).

The present study has several limitations that should be acknowledged. First, there are currently no clear data on the association between HPV infection (especially persistent HR-HPV infection) and key maternal-infant outcomes such as preterm labor or premature rupture of membranes; these data are being collected. Additionally, although traditional wet mount microscopy was employed for vaginal microecological assessment due to its clinical practicality (results available within 1 hour, enabling timely patient management), advanced techniques such as 16S rRNA sequencing or metagenomic analysis would provide more comprehensive insights into vaginal microbial composition, functional pathways, and metabolite profiles. We will incorporate advanced molecular detection methods in future studies. Third, the current single-center design with a limited sample size may restrict the generalizability of our findings. Future investigations should adopt a multicenter approach, enroll larger and age-stratified cohorts, and incorporate longitudinal follow-up.

In future research, a more comprehensive risk prediction model may be constructed by combining dynamic monitoring of HPV mRNA and accurate assessment of vaginal microecology during pregnancy. This will optimize the identification of high-risk pregnant women and facilitate individualized interventions and provide a basis for the development of a synergistic microecological-HPV prevention and control strategy to reduce the burden of HPV-related diseases during pregnancy and ensure the safety of mothers and infants.

Overall, integrating vaginal microecological assessment into routine prenatal care enables early detection of BV, facilitating timely intervention to reduce the risk of HR-HPV acquisition and persistence in pregnancy. Future studies should further explore how BV-associated biofilms and virulence factors disrupt the vaginal mucosal barrier and promote HPV infection.

## ACKNOWLEDGMENTS

We would like to thank the participants for their patience and kindness. This work was supported by Fujian Provincial Natural Science Foundation of China (Grant numbers 2023J011218 and 2022J011040).

## AUTHOR AFFILIATIONS

[1]Fujian Provincial Cervical Disease Diagnosis and Treatment Health Center, Fujian Maternity and Child Health Hospital College of Clinical Medicine for Obstetrics & Gynecology and Pediatrics Fujian Medical University, Fuzhou, Fujian, China

[2]Laboratory of Gynecologic Oncology, Fujian Maternity and Child Health Hospital College of Clinical Medicine for Obstetrics & Gynecology and Pediatrics, Fujian Medical University, Fuzhou, Fujian, China

[3]Fujian Key Laboratory of Women and Children's Critical Diseases Research, Fujian Maternity and Child Health Hospital (Fujian Women and Children's Hospital), Fuzhou, Fujian, China

[4]Fujian Clinical Research Center for Gynecological Oncology, Fujian Maternity and Child Health Hospital (Fujian Obstetrics and Gynecology Hospital), Fuzhou, Fujian, China

[5]Medical Genetic Diagnosis and Therapy Center, Fujian Maternity and Child Health Hospital College of Clinical Medicine for Obstetrics & Gynecology and Pediatrics, Fujian Key Laboratory for Prenatal Diagnosis and Birth Defect, Fujian Medical University, Fuzhou, Fujian, China

## AUTHOR ORCIDs

Jun Shen http://orcid.org/0000-0002-6795-164X
Liying Wang http://orcid.org/0009-0004-1679-3129
Jiancui Chen http://orcid.org/0000-0002-8960-1034
Pengming Sun http://orcid.org/0000-0002-5072-6091

## FUNDING

| Funder | Grant(s) | Author(s) |
|---|---|---|
| Fujian Provincial Natural Science Foundation of China | 2023J011218, 2022J011040 | Jun Shen |

## AUTHOR CONTRIBUTIONS

Jun Shen, Conceptualization, Data curation, Formal analysis, Funding acquisition, Investigation, Methodology, Project administration, Writing – original draft, Writing – review and editing | Liying Wang, Conceptualization, Data curation, Formal analysis, Funding acquisition, Investigation, Methodology, Writing – original draft, Writing – review and editing | Wenyu Lin, Conceptualization, Data curation, Formal analysis, Investigation, Methodology, Resources, Software, Supervision | Dingjie Wang, Formal analysis, Funding acquisition, Methodology, Project administration, Resources, Software, Supervision | Liang Wang, Formal analysis, Funding acquisition, Investigation, Methodology, Software | Jiancui Chen, Funding acquisition, Supervision, Validation, Visualization, Writing – original draft, Writing – review and editing | Pengming Sun, Funding acquisition, Project administration, Resources, Software, Supervision, Validation, Visualization, Writing – original draft, Writing – review and editing

## DATA AVAILABILITY

The data sets used and analyzed during the current study are available from the corresponding author on reasonable request.

## ETHICS APPROVAL

The study was approved by the Ethics Committee of Fujian Maternity and Child Health Hospital (Approval No. 2023KY027).

## ADDITIONAL FILES

The following material is available online.

Open Peer Review

**PEER REVIEW HISTORY (review-history.pdf).** An accounting of the reviewer comments and feedback.

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
