## [Reviewer comments · Microbiology Spectrum]

Microbiology Spectrum

Prevalence of vaginal microecological disorder and their association with HPV infections in pregnant women attending prenatal examinations

Jun Shen, Liying Wang, Wenyu Lin, Dingjie Wang, Liang Wang, Jiancui Chen, and Pengming Sun

Corresponding Author(s): Pengming Sun, Fujian Provincial Maternity and Children's Hospital

Review Timeline:

Submission Date:	August 7, 2025
Editorial Decision:	November 19, 2025
Revision Received:	February 20, 2026
Accepted:	March 10, 2026

Editor: Meghan Starolis

Reviewer(s): Disclosure of reviewer identity is with reference to reviewer comments included in decision letter(s). The following individuals involved in review of your submission have agreed to reveal their identity: Qun Wang (Reviewer #3)

Transaction Report:

DOI: <https://doi.org/10.1128/spectrum.02446-25>

Re: Spectrum02446-25 (Prevalence of vaginal microecological disorder and their association with HPV infections in pregnant women attending prenatal examinations)

Dear Dr. Pengming Sun:

Thank you for the privilege of reviewing your work. Below you will find my comments, instructions from the Spectrum editorial office, and the reviewer comments.

Revision Guidelines

Sincerely,
Meghan Starolis
Editor
Microbiology Spectrum

Reviewer #2 (Comments for the Author):

The authors describe the results of a cross-sectional study of 630 pregnant women receiving prenatal care at two hospitals over an approximately one-year period. High-risk HPV positives were identified using a clinically validated E6/E7 mRNA assay and results were correlated with vaginal microbiology results from swab samples using wet mounts and gram staining (Donders and Nugent scores). Leukocyte esterase (LE), neuraminidase 164 (SNA), hydrogen peroxide (H₂O₂) and pH were also measured. Diagnostic evaluations were then performed using the Vaginal Microecology Evaluation System. The authors found that vaginal

pH and Nugent score were associated with increased high-risk HPV infection and that BV-positive women were more likely to be infected with high-risk HPV. Taken together, this suggests that ecological imbalance results in increased risk of HPV infection. I found this to be a well written and clearly described study with objective endpoints and statistically supported conclusions. While the association of vaginal pathogens with high-risk infection has been described previously [Yang J, Long X, Li S, Zhou M and Hu L-N (2024) The correlation between vaginal pathogens and high-risk human papilloma virus infection: a meta-analysis of case control studies. *Front. Oncol.* 14:1423118. doi: 10.3389/fonc.2024.1423118 - not cited here], the authors correctly point out that pregnant women are understudied. This study confirms that these general findings also extend to pregnant women and that vaginal infections should be monitored during pregnancy.

I have the following questions for the authors' consideration:

1. You point out that HPV16/18/45 infections accounted for only 4.8% of high-risk infections with 23.8% infected with 11-other types - this suggests that there may be an impact of HPV vaccination in this population. Is vaccination status known or can it be inferred from regional data? Consider adding to discussion if known.
2. Another potential limitation of the study is the use of traditional wet mount techniques for vaginal microbiology combined with molecular HPV detection. Future studies might benefit from molecular detection of vaginal infections which are known to have higher sensitivity.
3. In the discussion section (paragraph beginning on line 257), the authors compare their findings to that of a systematic review from mainland China (reference 20) and suggest that their higher high-risk prevalence may be due to the use of an mRNA assay versus "conventional HPV DNA detection". They go on to speculate that this could be due to improved viral detection during pregnancy with an RNA-based assay, avoiding false positives due to latent or transient infections. I think that it is more likely these differences are due to the fact that the systematic review included routinely screened women, the majority of whom would not be pregnant, versus this cohort of exclusively pregnant women. E6/E7 mRNA is upregulated in cancer cases but the differences in pre-cancer (CIN2/3) are marginal at best (there were no cancer cases reported in this study). Moreover, in split sample testing, RNA and DNA assays perform very similarly.

Reviewer #3 (Comments for the Author):

Comment 1:

The text states that "AV had the highest prevalence of 33.1%," while Table 1 also reports AV as 33.1%. However, later in the text, it is stated that "AV was the most prevalent (12.7%, 80/630)." This discrepancy suggests that the 33.1% may represent the proportion among women with abnormal vaginal microecology, whereas the 12.7% reflects the proportion within the total study population. The authors should clarify these proportions explicitly to avoid confusion.

Comment 2:

In Figure 2 (forest plot), the reference categories are not clearly defined. It should be specified whether the comparisons are made against the HPV-negative group or the group with normal vaginal microecology. A precise description of the reference groups is essential for accurate interpretation of the results.

Comment 3:

The study does not address whether co-existing vaginal microecological imbalance further increases the risk of cervical cytological abnormalities (\geq ASC-US) among HPV-positive pregnant women. This analysis is critical for validating the hypothesis that vaginal microecological disorders act as synergistic risk factors for HPV infection. We recommend stratifying all HPV-positive women ($n=180$) by vaginal microecological status (normal vs. abnormal) and comparing the proportions of TCT results (\geq ASC-US (i.e., non-NILM) between the two groups.

Comment 4:

The study reports a clinically significant association between bacterial vaginosis (BV) prevalence and high-risk human papillomavirus (HR-HPV) infection status. However, while BV prevalence was significantly higher in women ≥ 35 years (25.6%) compared to younger pregnant women (9.0%, $P = 0.001$), the HR-HPV infection rate in the older group was significantly lower. This apparent contradiction warrants further statistical re-evaluation and discussion to clarify the underlying relationship.

Comment 5:

The reported rates of HPV positivity (28.6%) and vaginal microecological abnormalities (38.3%) are at the upper limit or exceed the ranges commonly reported in the literature. This suggests that the study population may not represent a general, unselected community-based cohort of pregnant women, but rather a clinically enriched or high-risk group. The authors should address potential selection bias and discuss the generalizability of their findings.

Comment 6:

The analysis of the association between BV and HPV infection (e.g., in Tables 3 and 5) relies solely on univariate methods such

as the chi-square test. However, age is significantly associated with both BV (Table 5) and HPV infection (Table 2), indicating that age is a potential confounder. Multivariate logistic regression analysis adjusting for age and other relevant factors is necessary to determine whether the BV-HPV association is independent. Without such adjustment, it remains unclear to what extent the observed association is genuine or attributable to confounding effects.

Spectrum02446-25

Prevalence of vaginal microecological disorder and their association with HPV infections in pregnant women attending prenatal examinations

Shen et al.

The authors describe the results of a cross-sectional study of 630 pregnant women receiving prenatal care at two hospitals over an approximately one-year period. High-risk HPV positives were identified using a clinically validated E6/E7 mRNA assay and results were correlated with vaginal microbiology results from swab samples using wet mounts and gram staining (Donders and Nugent scores). Leukocyte esterase (LE), neuraminidase 164 (SNA), hydrogen peroxide (H₂O₂) and pH were also measured. Diagnostic evaluations were then performed using the Vaginal Microecology Evaluation System. The authors found that vaginal pH and Nugent score were associated with increased high-risk HPV infection and that BV-positive women were more likely to be infected with high-risk HPV. Taken together, this suggests that ecological imbalance results in increased risk of HPV infection.

I found this to be a well written and clearly described study with objective endpoints and statistically supported conclusions. While the association of vaginal pathogens with high-risk infection has been described previously [Yang J, Long X, Li S, Zhou M and Hu L-N (2024) The correlation between vaginal pathogens and high-risk human papilloma virus infection: a meta-analysis of case control studies. *Front. Oncol.* 14:1423118. doi: 10.3389/fonc.2024.1423118 – not cited here], the authors correctly point out that pregnant women are understudied. This study confirm that these general findings also extend to pregnant women and that vaginal infections should be monitored during pregnancy.

I have the following questions for the authors' consideration:

1. You point out that HPV16/18/45 infections accounted for only 4.8% of high-risk infections with 23.8% infected with 11-other types – this suggests that there may be an impact of HPV vaccination in this population. Is vaccination status known or can it be inferred from regional data? Consider adding to discussion if known.
2. Another potential limitation of the study is the use of traditional wet mount techniques for vaginal microbiology combined with molecular HPV detection. Future studies might benefit from molecular detection of vaginal infections which are known to have higher sensitivity.
3. In the discussion section (paragraph beginning on line 257), the authors compare their findings to that of a systematic review from mainland China (reference 20)

and suggest that their higher high-risk prevalence may be due to the use of an mRNA assay versus “conventional HPV DNA detection”. They go on to speculate that this could be due improved viral detection during pregnancy with an RNA-based assay, avoiding false positives due to latent or transient infections. I think that it is more likely these differences are due to the fact that the systematic review included routinely screened women, the majority of whom would not be pregnant, versus this cohort of exclusively pregnant women. E6/E7 mRNA is upregulated in cancer cases but the differences in pre-cancer (CIN2/3) are marginal at best (there were no cancer cases reported in this study). Moreover, in split sample testing, RNA and DNA assays perform very similarly.

Manuscript Evaluation Form

Microbiology Spectrum

Manuscript Title: “Prevalence of vaginal microecological disorder and their association with HPV infections in pregnant women attending prenatal examinations”.

Comments to the Editor:

I have reviewed the manuscript and hereby submit my comments for your consideration:

Comment 1:

The text states that "AV had the highest prevalence of 33.1%," while Table 1 also reports AV as 33.1%. However, later in the text, it is stated that "AV was the most prevalent (12.7%, 80/630)." This discrepancy suggests that the 33.1% may represent the proportion among women with abnormal vaginal microecology, whereas the 12.7% reflects the proportion within the total study population. The authors should clarify these proportions explicitly to avoid confusion.

Comment 2:

In Figure 2 (forest plot), the reference categories are not clearly defined. It should be specified whether the comparisons are made against the HPV-negative group or the group with normal vaginal microecology. A precise description of the reference groups is essential for accurate interpretation of the results.

Comment 3:

The study does not address whether co-existing vaginal microecological imbalance further increases the risk of cervical cytological abnormalities (\geq ASC-US) among HPV-positive pregnant women. This analysis is critical for validating the hypothesis that vaginal microecological disorders act as synergistic risk factors for HPV infection. We recommend stratifying all HPV-positive women ($n=180$) by vaginal microecological status (normal vs. abnormal) and comparing the proportions of TCT results \geq ASC-US (i.e., non-NILM) between the two groups.

Comment 4:

The study reports a clinically significant association between bacterial vaginosis (BV) prevalence and high-risk human papillomavirus (HR-HPV) infection status. However, while BV prevalence was significantly higher in women ≥ 35 years (25.6%) compared to younger pregnant women (9.0%, $P = 0.001$), the HR-HPV infection rate in the older group was significantly lower. This apparent contradiction warrants further statistical re-evaluation and discussion to clarify the underlying relationship.

Comment 5:

The reported rates of HPV positivity (28.6%) and vaginal microecological abnormalities (38.3%) are at the upper limit or exceed the ranges commonly reported in the literature. This suggests that the study population may not represent a general, unselected community-based

cohort of pregnant women, but rather a clinically enriched or high-risk group. The authors should address potential selection bias and discuss the generalizability of their findings.

Comment6:

The analysis of the association between BV and HPV infection (e.g., in Tables 3 and 5) relies solely on univariate methods such as the chi-square test. However, age is significantly associated with both BV (Table 5) and HPV infection (Table 2), indicating that age is a potential confounder. Multivariate logistic regression analysis adjusting for age and other relevant factors is necessary to determine whether the BV - HPV association is independent. Without such adjustment, it remains unclear to what extent the observed association is genuine or attributable to confounding effects.

福建省妇幼保健院·福建医科大学妇儿临床医学院

Fujian Maternity and Child Health Hospital
College of Clinical Medicine for Obstetrics & Gynecology and Pediatrics, Fujian Medical University

Editor-in-Chief

Dr. Christina Cuomo

Microbiology Spectrum

Dear Editors and Reviewers:

Thank you for your letter and for the reviewer's comments concerning our manuscript entitled " **Prevalence of vaginal microecological disorder and their association with HPV infections in pregnant women attending prenatal examinations**". (Submission ID: Spectrum02446-25). These comments are all valuable and very helpful for revising and improving our paper, as well as the important guiding significance to our researches. We have studied comments carefully and have made corrections which we hope meet with approval. The main corrections in the paper and the responds to the reviewer's comments areas flowing:

Responses to the reviewer's comments:

Reviewer #2 (Comments for the Author):

The authors describe the results of a cross-sectional study of 630 pregnant women receiving prenatal care at two hospitals over an approximately one-year period. High-risk HPV positives were identified using a clinically validated E6/E7 mRNA assay and results were correlated with vaginal microbiology results from swab samples using wet mounts and gram staining (Donders and Nugent scores). Leukocyte esterase (LE), neuraminidase 164 (SNA), hydrogen peroxide (H₂O₂) and pH were also measured. Diagnostic evaluations were then performed using the Vaginal Microecology Evaluation System. The authors found that vaginal pH and Nugent score were associated with increased high-risk HPV infection and that BV-positive women were more likely to be infected with high-risk HPV. Taken together, this suggests that ecological imbalance results in increased risk of HPV infection.

I found this to be a well written and clearly described study with objective endpoints and statistically supported conclusions. While the association of vaginal pathogens with high-risk infection has been described previously [Yang J, Long X, Li S, Zhou M and Hu L-N (2024) *The correlation between vaginal pathogens and high-risk human papilloma virus infection: a meta-analysis of case control studies. Front. Oncol. 14:1423118. doi: 10.3389/fonc.2024.1423118 - not cited here*], the authors correctly

福建省妇幼保健院·福建医科大学妇儿临床医学院

Fujian Maternity and Child Health Hospital
College of Clinical Medicine for Obstetrics & Gynecology and Pediatrics, Fujian Medical University

point out that pregnant women are understudied. This study confirm that these general findings also extend to pregnant women and that vaginal infections should be monitored during pregnancy.

Response: Thank you for your valuable comments. We have cited Yang et al. (2024) as Reference 15 to clarify that vaginal microecological imbalance increases HR-HPV infection risk (Page 5, Lines 94-97). Given the limited research on vaginal microecology in pregnant women, this study investigates the prevalence of such disorders and their association with HPV infections in this understudied population, while emphasizing the clinical need for monitoring vaginal infections during pregnancy.

I have the following questions for the authors' consideration:

1. You point out that HPV16/18/45 infections accounted for only 4.8% of high-risk infections with 23.8% infected with 11-other types - this suggests that there may be an impact of HPV vaccination in this population. Is vaccination status known or can it be inferred from regional data? Consider adding to discussion if known.

Response: Following your suggestion, we collected HPV vaccination status via follow-up telephone interviews and questionnaires. Among participants, 533 (84.6%) were unvaccinated, while 97 (15.4%) had received HPV vaccines (12 [1.9%] bivalent, 42 [6.7%] quadrivalent, 43 [6.8%] 9-valent; **Table 1**). We further added analyses of the association between vaccination status and HPV infection in **Tables 3 and 5**, demonstrating a significantly reduced HR-HPV risk in vaccinated individuals($P=0.003$).

In the Discussion (page 11 Lines 262-265), we clarified that HPV 52/58 were the most prevalent genotypes in Fujian, with HPV 16/18 prevalence at 1.74%. These epidemiological data reflect the long-term efficacy of HPV vaccination, underscoring the need to further expand access to the HPV vaccine.

2. Another potential limitation of the study is the use of traditional wet mount techniques for vaginal microbiology combined with molecular HPV detection. Future studies might benefit from molecular detection of vaginal infections which are known to have higher sensitivity.

Response: We acknowledge this limitation and have added a discussion (page 13 Lines 339-347). While molecular techniques offer higher sensitivity, traditional wet mount microscopy was chosen for rapid clinical results (within one hour), enabling timely treatment. Future studies will consider molecular methods.

福建省妇幼保健院·福建医科大学妇儿临床医学院

Fujian Maternity and Child Health Hospital
College of Clinical Medicine for Obstetrics & Gynecology and Pediatrics, Fujian Medical University

3. In the discussion section (paragraph beginning on line 257), the authors compare their findings to that of a systematic review from mainland China (reference 20) and suggest that their higher high-risk prevalence may be due to the use of an mRNA assay versus "conventional HPV DNA detection". They go on to speculate that this could be due improved viral detection during pregnancy with an RNA-based assay, avoiding false positives due to latent or transient infections. I think that it is more likely these differences are due to the fact that the systematic review included routinely screened women, the majority of whom would not be pregnant, versus this cohort of exclusively pregnant women. E6/E7 mRNA is upregulated in cancer cases but the differences in pre-cancer (CIN2/3) are marginal at best (there were no cancer cases reported in this study). Moreover, in split sample testing, RNA and DNA assays perform very similarly.

Response: We thank you for this insight and have revised the Discussion (page 12 Lines 285-296). As supported by literature, HR-HPV prevalence in pregnancy varies widely (5.5% – 65%) due to age, geography, and gestational age. Prior studies show no significant difference in CIN II/III detection rates between HR-HPV E6/E7 mRNA and DNA assays. Our cohort's higher prevalence may reflect its composition (pregnant women from a tertiary hospital). Future multicenter studies with community populations and age stratification are warranted.

Reviewer #3 (Comments for the Author):

Comment 1:

The text states that "AV had the highest prevalence of 33.1%," while Table 1 also reports AV as 33.1%. However, later in the text, it is stated that "AV was the most prevalent (12.7%, 80/630)." This discrepancy suggests that the 33.1% may represent the proportion among women with abnormal vaginal microecology, whereas the 12.7% reflects the proportion within the total study population. The authors should clarify these proportions explicitly to avoid confusion.

Response: We have revised the text and **Table 1** to uniformly report AV prevalence as 12.7% (80/630) of the total study population (N=630). We appreciate your careful review.

Comment 2:

In Figure 2 (forest plot), the reference categories are not clearly defined. It should be specified whether the comparisons are made against the HPV-negative group or the group with normal vaginal microecology. A precise description of the reference

福建省妇幼保健院·福建医科大学妇儿临床医学院

Fujian Maternity and Child Health Hospital
College of Clinical Medicine for Obstetrics & Gynecology and Pediatrics, Fujian Medical University

groups is essential for accurate interpretation of the results.

Response: Thank you for your comment. We have removed **Figure 2**. The reference groups and detailed results are clearly specified and presented in **Tables 3**.

Comment 3:

The study does not address whether co-existing vaginal microecological imbalance further increases the risk of cervical cytological abnormalities (\geq ASC-US) among HPV-positive pregnant women. This analysis is critical for validating the hypothesis that vaginal microecological disorders act as synergistic risk factors for HPV infection. We recommend stratifying all HPV-positive women ($n=180$) by vaginal microecological status (normal vs. abnormal) and comparing the proportions of TCT results \geq ASC-US (i.e., non-NILM) between the two groups.

Response: As recommended, we added **Table 4**, stratifying 180 HPV-positive women by vaginal microecology status and TCT results (page 10 Lines 228-232).

Comment4:

The study reports a clinically significant association between bacterial vaginosis (BV) prevalence and high-risk human papillomavirus (HR-HPV) infection status. However, while BV prevalence was significantly higher in women ≥ 35 years (25.6%) compared to younger pregnant women (9.0%, $P = 0.001$), the HR-HPV infection rate in the older group was significantly lower. This apparent contradiction warrants further statistical re-evaluation and discussion to clarify the underlying relationship.

Response: Our re-evaluation (**Table 6**) shows that among HR-HPV-positive pregnant women, BV remains significantly associated with HR-HPV infection after age adjustment (OR = 2.43; $P=0.009$), confirming BV as an independent risk factor.

Comment5:

The reported rates of HPV positivity (28.6%) and vaginal microecological abnormalities (38.3%) are at the upper limit or exceed the ranges commonly reported in the literature. This suggests that the study population may not represent a general, unselected community-based cohort of pregnant women, but rather a clinically enriched or high-risk group. The authors should address potential selection bias and discuss the generalizability of their findings.

Response: Thank you for your comment. We added literature to the Discussion noting that HR-HPV prevalence in pregnancy ranges from 5.5% to 65% (page 11

福建省妇幼保健院·福建医科大学妇儿临床医学院

Fujian Maternity and Child Health Hospital
College of Clinical Medicine for Obstetrics & Gynecology and Pediatrics, Fujian Medical University

lines 287-290). A third limitation addressing selection bias was included (Page 14, Lines 345-347), highlighting the need for multicenter, age-stratified studies.

Comment6:

The analysis of the association between BV and HPV infection (e.g., in Tables 3 and 5) relies solely on univariate methods such as the chi-square test. However, age is significantly associated with both BV (Table 5) and HPV infection (Table 2), indicating that age is a potential confounder. Multivariate logistic regression analysis adjusting for age and other relevant factors is necessary to determine whether the BV-HPV association is independent. Without such adjustment, it remains unclear to what extent the observed association is genuine or attributable to confounding effects.

Response: We performed age-adjusted multivariate logistic regression for **Tables 3 and 5**. BV, Nugent score, and leukocyte esterase positivity remained significant risk factors post-adjustment.

Prof. Dr. Pengming Sun

Chief Physician

Deputy Superintendent

Fujian Maternity and Child Health Hospital

College of Clinical Medicine for Obstetrics & Gynecology and Pediatrics, Fujian medical University

No. 18, Daoshan Road, Gulou District, Fuzhou, 350001, Fujian, P.R. of China

Tel: 0086-591-87558732;

Mobile: 0086-13788873900

Fax: 0086-591-87551247

E-mail: fmsun1975@fjmu.edu.cn / sunfemy@hotmail.com

Re: Spectrum02446-25R1 (Prevalence of vaginal microecological disorder and their association with HPV infections in pregnant women attending prenatal examinations)

Dear Dr. Pengming Sun:

Your manuscript has been accepted, and I am forwarding it to the ASM production staff for publication. Your paper will first be checked to make sure all elements meet the technical requirements. ASM staff will contact you if anything needs to be revised before copyediting and production can begin. Otherwise, you will be notified when your proofs are ready to be viewed.

Sincerely,
Meghan Starolis
Editor
Microbiology Spectrum